# Primate Erythroparvovirus 1 Infection in Patients with Hematological Disorders

**DOI:** 10.3390/pathogens11050497

**Published:** 2022-04-21

**Authors:** Stefka Krumova, Ivona Andonova, Radostina Stefanova, Polina Miteva, Galina Nenkova, Judith M. Hübschen

**Affiliations:** 1National Reference Laboratory “Measles, Mumps, Rubella”, Department of Virology, National Center of Infectious and Parasitic Diseases, 1233 Sofia, Bulgaria; ivona_a@yahoo.com (I.A.); rss_94@abv.bg (R.S.); 2Specialized Hospital for Active Treatment of Children’s Diseases “Prof. Dr. Ivan Mitev” EAD, 1612 Sofia, Bulgaria; pmiteva2020@gmail.com; 3Department of Biology, Medical University, 9002 Varna, Bulgaria; g_alina_n@abv.bg; 4Clinical and Applied Virology Group, Department of Infection and Immunity, Luxembourg Institute of Health, 4354 Esch-sur-Alzette, Luxembourg; judith.huebschen@lih.lu

**Keywords:** Parvovirus B19 (B19V), anemia, IgM antibodies, viral DNA, kidney recipients

## Abstract

Primate erythroparvovirus 1, commonly referred to as Parvovirus B19 (B19V), is a DNA virus that normally results in a mild childhood infection called “erythema infectiosum”. Besides respiratory spread, B19V can also be transmitted through transfusions, which may result in persistent anemia in immunodeficient hosts. Dialysis patients often face acute or chronic anemia after infection with B19V. Here, we describe the laboratory investigation of 21 patients with hematological disorders for B19V infections. B19V DNA was detected in 13 (62%) of them, with specific IgM antibodies in three of the DNA positives. All 13 patients received treatment and were laboratory-monitored over a period of one year. In only two patients (a 14-year-old child with a kidney transplantation and a 39-year-old patient with aplastic anemia), markers of recent B19V infection were still detectable in follow-up samples. For four B19V DNA positive samples, short sequences could be obtained, which clustered with genotype 1a reference strains. Our findings suggest that all cases of hematological disorders should be examined for specific B19V antibodies and DNA for accurate diagnosis and appropriate patient management.

## 1. Introduction

Primate erythroparvovirus 1, commonly referred to as Parvovirus B19 (B19V), is a nonenveloped, single-stranded DNA virus that is 18–26 nm in diameter [1]. B19V is a human pathogen with global distribution and is generally transmitted via respiratory droplets [2]. The lack of an envelope makes the virus quite resistant to physical and chemical inactivation as well as treatment with detergents, and it often survives in blood products [3,4]. Viral replication takes place in human bone marrow cells and shows a pronounced tropism to erythroid precursors. The main virus receptor is the blood group P antigen, which is on the surface of a wide range of cells, including erythroblasts, megakaryocytes, endothelial cells, fetal myocytes, hepatocytes, placental trophoblasts, and others [5].

Clinically, B19V is a pathogen that is associated with a wide range of disease manifestations, syndromes, and pathological conditions with acute or chronic course, some of which depend largely on the condition of the immune system and the hematological status of the patient. B19V may cause several acute diseases such as “erythema infectiosum” in children or acute polyarthritis in adults, as well as transient aplastic crisis in immunocompetent hosts [6,7]; in addition, immunocompromised patients may face persistent infection, leading to chronic anemia or, more rarely, thrombocytopenia, neutropenia, and pancytopenia [8,9,10]. As recently reported, renal transplant recipients may experience glomerulopathy and allograft dysfunction, especially after primary infections through the transplanted organ. Since infections may result in various clinical pictures, which complicate diagnosis, it was suggested that hematologic disorders without clear clinical diagnosis should be investigated for B19V [11]. However, testing for specific IgM antibodies is often not sufficient to establish a B19V diagnosis in immunocompromised patients. Virus detection relies on polymerase chain reaction (PCR) in serum or tissue samples. In case of positivity, treatment with intravenous anti-B19V-specific immunoglobulin (Ig) often leads to an excellent clinical and virological response [9,12].

The aim of the work described here was to investigate whether B19V plays a role in the development of hematological disorders in immunocompetent individuals and in kidney transplant children in Bulgaria, using different laboratory diagnostic tools.

## 2. Results

### 2.1. Detection of B19V in Patients with Different Clinical Diagnoses

In the frame of routine patient investigation, we screened samples from 21 patients with hematological disorders (anemia, thrombocytopenia, and thalassemia major), including two kidney recipients with anemia for B19V. Three of the 21 patients (14%) were IgM positive and all of them had been diagnosed with aplastic anemia (Table 1). Specific anti-B19V IgG antibodies were found in 19 (90%) patients.

B19V DNA was detected in 13 (62%) samples from 10 patients with aplastic anemia and one each with thrombocytopenia, Thalassemia major, and renal transplantation combined with anemia using the NS1-PCR (Table 1), while there were no positives in the VP1u-PCR or the nested PCR. Two of the patients with detectable B19V DNA were positive only by PCR without a proven other viral marker. The three anti-B19V IgM positive patients were also positive for IgG antibodies and B19V DNA. Eight patients exhibited a combination of anti-B19V IgG antibodies and B19V DNA.

### 2.2. Sequencing and Phylogenetic Analysis

All B19V DNA positive samples were submitted for sequencing. The four sequences obtained from the three IgM positive patients and one patient with kidney transplantation and negative IgM clustered with genotype 1a reference sequences (Figure 1).

All patients with markers of acute B19V infection were laboratory-monitored over a period of one year. In a 39-year-old patient with aplastic anemia, B19V DNA and anti-B19V IgM antibodies were still detectable three months after primary clinical symptoms. In one of the kidney transplantation patients, B19V infection was confirmed by PCR two months after transplantation and in a follow-up sample three months after transplantation. Six months after transplantation, B19V DNA was no longer detectable and only IgG antibodies persisted. The second child with kidney transplantation did not show any signs of recent B19V infection. The follow-up samples of all other patients with detectable DNA at the first test were negative for B19V DNA and IgM antibodies, but positive for IgG antibodies two months after the initial test.

All patients with markers of recent B19V infection were treated with corticosteroids, intravenous immunoglobulin, paracetamol, and infusions of aqueous saline solutions, with good outcomes of the infection being observed.

## 3. Discussion

B19V has a marked tropism for erythroid progenitor cells. This determines its pathology and the development of chronic anemia and disorders of hematopoiesis in immunocompromised individuals [5]. An important clinical aspect is the risk of infection through B19V-contaminated blood products. An earlier report [13] found a high rate of B19V seroprevalence in patients having received multiple blood transfusions or blood products and a prospective study found evidence of B19V infection in 13/43 (30%) patients with chronic anemia [14]. The precise selection of donors and donor products, including the use of nucleic acid testing (NAT), reduces the risk of spreading blood-borne infectious agents. B19 is a proven risk factor in transfusion medicine. Previous studies [15,16] have shown the presence of B19V DNA in 0.4% to 1.4% of blood donors; on the other hand, another study [17] found more than 50% B19V IgG seroprevalence in Australian blood donors. We documented active B19V infection in 13 of 21 patients with hematological disorders (62%) and IgG antibodies in 19 (90%) of them. The latter could be due to previous virus exposure or transfusion of blood products containing these antibodies.

After the first report of B19V infection following transplantation [18], many cases of B19V infections after solid-organ or hematopoietic stem cell transplantation were published, with anemia being the most frequent clinical manifestation [19]. Renal transplant patients are not routinely screened for B19V infection and often immunosuppressive drugs are held responsible for symptoms such as anemia and/or pancytopenia. B19V screening one month after transplantation and based on clinical suspicion may help to clarify the role of B19V infection in these patients [2]. The average time between transplantation and the appearance of symptomatic anemia was 8.6 weeks in a previous study [20]. In our report, we described two cases of children with anemia and a recent kidney transplantation and in one of them; a B19V infection was proven and the symptoms were present with anemia 7 weeks after transplantation. After treatment, the infection and anemia were controlled. We emphasize the importance of considering B19V infection in patients with anemia after transplantation, especially if high doses of immunosuppressants are administered, since early and targeted treatment can prevent the worsening of clinical symptoms with possible consequences on transplantation success.

Confirmation of active B19V infection can, in principle, be done by the detection of viral DNA or specific IgM antibodies. However, IgM assay results are less reliable due to delayed or inadequate antibody responses in immunosuppressed individuals [13]. Viral DNA can be detected in various clinical specimens, as well as in asymptomatic individuals. However, viral DNA detection in combination with clinical symptoms is likely to represent an active infection, as shown in the present study where 13 patients were PCR positive, but only three had specific IgM antibodies. The NS1-PCR was used for the initial molecular diagnosis with high sensitivity to different B19V genetic variants, while the other two PCRs were done later on NS1-PCR positive samples and after spotting of the material on FTA cards. Thus, suboptimal recovery of material from the FTA cards, DNA degradation, and the previously observed lower sensitivity of the other two PCRs [21] might explain why only the NS1-PCR yielded positive results.

Due to the relatively small number of samples with sequence information, we cannot link the B19V genotype and hematological manifestations of infection. The four successfully sequenced samples clustered with genotype 1a strains and, although the obtained sequence information covered only 63 nucleotides, this result is in agreement with a previous study reporting that genotype 1a has been dominant in Bulgaria for a very long time [21]. There do not seem to be disease-specific B19V genotypes, but some authors reported a predominance of genotype 1 in immuno-compromised patients [22] and in blood donors [16,23]. Previous research [24] has shown that the most common infection of patients with hematological diseases with B19V is subgenotypes 1a and 3b, and the spread of genotype 2 is almost disappearing. The study of genotypic identity of patients infected with B19V is important because of the risk of co-infection between different genotypes, especially in immunocompromised patients [24].

## 4. Materials and Methods

### 4.1. Patients

Serum samples were tested from 21 patients between 10 and 39 years of age (average 17 years) with hematological diagnoses of aplastic anemia (*n* = 13), thrombocytopenia (4), transfusion-dependent anemia (thalassemia major) (*n* = 2), and renal transplant recipients with anemia (*n* = 2). All patients had received between one and three blood transfusions, at least one of them in the year of serum collection.

The B19V laboratory examination of clinical materials from patients with hematologic diseases described here was part of the routine screening of these patients for infectious agents that are associated with hematological disorders. Clinical specimens were sent for testing to the National Reference Laboratory “Measles, Mumps, Rubella”, National Centre of Infectious and Parasitic Diseases, Sofia, Bulgaria from hematology departments of different hospitals in the country.

Routine laboratory investigations were carried out, including complete blood picture, as part of patient management. The laboratory results obtained were important for patient treatment decisions.

B19V infection was classified as recent in patients with detectable viral DNA and/or specific IgM antibodies, and as old in patients with only IgG antibodies. The patients’ specimens were tested in parallel by ELISA and PCR and follow-up samples were requested from all patients with recent B19V infection, regardless of the identified laboratory marker.

### 4.2. ELISA Assays

To detect specific B19V IgM and IgG antibodies, commercial indirect ELISA assays were used (Euroimmun, Lübeck, Germany) according to the manufacturer’s instructions. The results were interpreted as positive, negative, and equivocal.

### 4.3. PCR Assays and Sequencing

Viral DNA extraction was done from all serum samples using the NucleoSpin Blood test kits (Macherey-Nagel GmbH & Co. KG, Duren, Germany). Screening for B19V DNA was performed by AmpliTaq Gold kits (Thermo Fisher Scientific, Inc., Waltham, MA, USA) and with primers e1905f and e1987r located in the NS1 gene (NS1-PCR), primers e2717f and e2901r targeting the VP1 unique region (VP1u-PCR), and nested-PCR spanning the NS1/VP1u junction [25]. All PCRs contained negative and positive controls and the products were analysed in 2% agarose gels stained with ethidium bromide. PCR products of the expected size were purified with the QIAquick PCR purification kits (Qiagen) and sequenced with the BigDye Terminator v3.1 cycle sequencing kit (Life Technologies, Carlsbad, CA, USA). Sequences were edited with SeqScape v2.5 and aligned with reference strains in BioEdit v7.1. Phylogenetic analysis was done with MEGA7 [26] using the Kimura 2-parameter and the Neighbor-Joining methods.

## 5. Conclusions

B19V is a proven pathological factor in a number of hematological diseases related to frequent blood transfusions and may manifest as refractory anemia during the post-transplantation period. Viral DNA detection helps to establish diagnosis, especially in immunosuppressed patients with an impaired humoral response following B19V infection. Our findings suggested that all cases of hematological disorders should be examined for specific B19V antibodies and DNA for accurate diagnosis and appropriate patient management.

## Figures and Tables

**Figure 1 pathogens-11-00497-f001:**
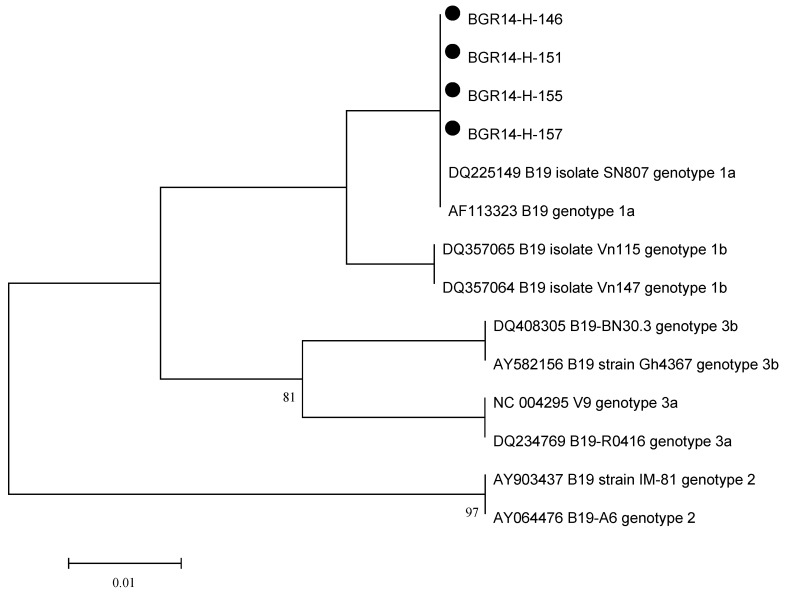
Phylogenetic tree based on an alignment covering 63 nucleotides of the B19V NS1 gene and the Kimura 2-parameter and Neighbor-Joining methods. Bootstrap values of at least 70 (based on 1000 resamplings) are shown at the nodes, with sequences from Bulgaria being marked with a black dot.

**Table 1 pathogens-11-00497-t001:** B19V test results by clinical diagnosis.

Clinical Diagnosis	Tested Patients	No B19V ELISA IgM Positive	No B19V ELISA IgG Positive	No B19V PCR NS1 Region Positive	Genotyping (NS1 Region)
Aplastic anemia	13	3	11	10	3 (1a)
Thrombocytopenia	4	-	4	1	
Thalassemia major	2	-	2	1	
Renal transplant recipients with anemia	2	-	2	1	1 (1a)
**Total**	**21**	**3**	**19**	**13**	**4 (1a)**

## Data Availability

Not applicable.

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
