# Peer review of "Primate Erythroparvovirus 1 Infection in Patients with Hematological Disorders"

_pathogens, 2022, doi:10.3390/pathogens11050497_

Round 1
Reviewer 1 Report
References from the last five years were not used in the article. The article should be supported by up-to-date information using a few references from the last five years.I believe that using up-to-date references in the introduction and discussion section will make the study more valuable.
Author Response
Response to Reviewer Comments
Point 1: References from the last five years were not used in the article. The article should be supported by up-to-date information using a few references from the last five years.
I believe that using up-to-date references in the introduction and discussion section will make the study more valuable.
Response: Dear reviewer, thank you for your comments. The necessary revised, according to your recommendation, have been made in the manuscript: corrections on line 118-122 (included text) and included references: №10, 15, 16, 17 and 24 (in red)
Please, see attached files.

Reviewer 2 Report
Krumova et al. has investigated the incidence of Parvovirus B19 (B19V) infections in the patients of hematological disorders using different diagnostic methods. Moreover, this study emphasizes the detection of B19 specific antibodies and viral DNA as a marker of initial diagnosis especially in the cases of hematological disorders. There are minor concerns that needs to be addressed before final consideration.
Author has detected 13 positive cases of B19 and only 4 samples were sequenced for genotyping. Why were the rest of the samples not sequenced?
This study reported a genotype 1a of B19 virus in studied cases. What is the significance of genotyping B19 positive samples in this case? Does specific genotype correlate with the disease severity? Please discuss it in the discussion.
This study was performed on clinical samples. Author should mention the ethical statement in the methods section.
Author Response
Response to Reviewer 2 Comments
Point 1 Author has detected 13 positive cases of B19 and only 4 samples were sequenced for genotyping. Why were the rest of the samples not sequenced?
Response 1: Dear reviewer, thank you for your comments. In the present study, all B19V DNA PCR positive samples (n=13) were sequenced, but only from four of them were obtained successful sequences with the possibility of phylogenetic analysis. For the other samples, very short sequences were obtained and they were excluded from the analysis. This may be due to the storage of samples on FTA cards as well as low viral load. B19V NS1/VP1u junction region is used as the main approach for B19V phylogenetic analysis, but as described in line 76-79 and line 144-165, in the present study, positive PCR results were obtained only for the diagnostic and highly conserved B19V-NS1 region. (in red)
Point 2 This study reported a genotype 1a of B19 virus in studied cases. What is the significance of genotyping B19 positive samples in this case? Does specific genotype correlate with the disease severity? Please discuss it in the discussion.
Response 2: Dear reviewer, thank you for your comments. The corrections have been made and comments have been included in the discussion (line 167-172) and included reference №25. (in red)
Point 3 This study was performed on clinical samples. Author should mention the ethical statement in the methods section.
Response 3: Dear reviewer, thank you for your comments. This has been corrected and included in line 214-216 from the manuscript. (in red)
The necessary revised, according to your recommendations, have been made in the manuscript. Please, see attached file.
